# Teachers' Perception towards the Use of Quizizz in the Teaching and Learning of English: A Systematic Review

**Thomas Mason Lim** *,† and **Melor Md Yunus** †

Faculty of Education, Universiti Kebangsaan Malaysia, Bangi 43600, Selangor, Malaysia; melor@ukm.edu.my
* Correspondence: tommylim5353@yahoo.com; Tel.: +60-111-149-8535
† Both authors contributed equally to this work.

**Abstract:** The teaching and learning of English as a second language have always been emphasized by the Ministry of Education (MOE). In the Malaysia Education Blueprint (2013–2025), among the many goals is to produce learners who can comprehend the language for various purposes, including for information and enjoyment. Over the years, the teaching and learning of English have changed from conventional chalk-and-talk methods to modern methods that involve various platforms such as Quizizz, which is free and easy to use. This systematic literature review focuses on teachers' perception on the use of Quizizz in the teaching and learning of English. Using the Preferred Reporting Items for Systematic Review and Meta-Analyses (PRISMA) review methodology, a total of 45 articles related to the use of Quizizz in the English language classroom were identified from the ERIC and Google Scholar databases. Based on the articles from the year 2017 to 2021, the results show that the mixed-method research design was most used to gather teachers' perspectives on the use of Quizizz in the teaching and learning of English. Overall, the results of this study show that Quizizz is accepted positively among teachers due to its effectiveness, feasibility, ease of use, and motivating nature for learners. As a suggestion for future research, more focus can be put on investigating the effectiveness of Quizizz from the perspective of parents and issues related to the implementation of gamified learning, such as Internet connection and device availability.

**Keywords:** education; Quizizz; systematic literature review; teachers' perspective; game-based learning; PRISMA

## 1. Introduction

The trend of using online platforms for education has taken the world by storm, as many have come to realize the potential these platforms hold in providing better education for learners of all levels. Researchers have argued that e-learning makes learning more enjoyable, aside from providing a multitude of advantages. Furthermore, e-learning disseminates information in a different yet effective and long-lasting manner [1]. In recent years, studies have proven that e-learning is more alluring for learners because it motivates them to learn, provides an interactive learning environment, and affords learners the opportunity to learn collaboratively and meaningfully. Furthermore, e-learning is in fact beneficial in allowing for immediate feedback in context, aside from being able to promote learning even among anxious learners. In a research conducted by Rubio-Valdehita et al. [2], they argued that it is strongly believed that e-learning is widely accepted by users of various backgrounds because technology caters to every type of human ability, be it weak or strong. The source of motivation to conduct this study came from the home-based-learning (HBL) period from the beginning of the Movement Control Order (MCO) due to COVID-19 back in 2020. Researchers saw a rise in the use of Quizizz among teachers, and it became a question of whether the platform is effective in improving learners' academic performance and their knowledge development. The teachers' perception can be predicted but a review of articles based on the use of Quizizz turned out to be an effective method to truly

understand teachers' perception of the aforementioned online learning platform. Therefore, this systematic literature review was carried out to investigate teachers' perspective on the use of e-learning, specifically Quizizz, in the teaching and learning of the English language. It is hoped that the findings from this systematic literature review will shine more light onto the use, effectiveness, and feasibility of Quizizz, specifically among teachers and various stakeholders of the education system across the world.

In the following sections, the purpose of this review will be looked into in detail, and it will then be followed by the step-by-step methodology of conducting this review using the PRISMA 2020 checklist. From there onwards, the findings from the articles reviewed and a conclusion to answer the research objectives of this study will be discussed. To begin, a deeper insight into the teaching and learning of the English language and game-based learning will be considered in the following subsections.

## 1.1. Teaching and Learning of English

It has become a norm everywhere for people to communicate using the English language, as it is a common lingua franca [3]. The teaching and learning of the English language for communication does not only happen in one country but rather is happening all over the world [4]. The four skills that are heavily focused on in the teaching and learning of the English language are listening, speaking, reading, and writing. The teaching and learning of English can be broken down into several types, such as English as a foreign language (EFL), English as a lingua franca (ELF), and English as a second language (ESL). In Malaysia, English is perceived as the nation's second language. Therefore, learners in Malaysia are indirectly required to have at least the most basic command of the English language from primary school. The Ministry of Education (MOE) sees the English language as one of the most important languages because it acts as the language of various affairs worldwide. As asserted by Qin [5], aside from being the language of business and communication, the English language is also important for one's employability when entering the workforce.

Gearing towards the Fourth Industrial Revolution (IR 4.0), English is now even more important than ever due to the demand for proficient English language speakers from most employers [6]. A potential employee's ability to communicate well in the English language plays a huge part in getting them employed in this fast-paced world, where competition between different companies and entities is continuously happening every day. Considering this, the MOE is invested in making sure that teachers and learners are always up to date with the latest teaching and learning methods that can be used and accessed inside and outside of the classroom [7]. It is also a major goal of the MOE to make sure that all learners can use the language at its optimum level so that they can communicate effectively in the future global competitive economy [8].

However, despite the various efforts taken by the MOE alongside non-governmental organizations (NGOs) at various levels, students' level of proficiency in the English language is still low. This has led to many studies conducted to investigate the factors that cause students' level of proficiency to not improve. There were also many studies conducted to discover the best ways to teach the language to learners. It was discovered that one of the best methods, which most researchers would agree upon, is the use of technology in the teaching and learning of English. In a research conducted by Love [9], he discovered that technology serves as a productive approach in the mission to achieve strong results in terms of learners' level of proficiency in the English language.

Interactive learning is highly accepted these days, especially by learners, because they have more fun during their lessons. Although it is more common in higher-level education, interactive learning is, in fact, beneficial for all levels of education, including primary and secondary education [10]. Commonly known as e-learning, it has proven to increase students' motivation towards and engagement with their lessons. After much research was conducted, claims were made by researchers about how digital learning in the classroom can bring benefits that are in accordance with the teaching and learning of

English. For instance, digital gaming encourages self-discovery where teachers are not needed to direct the students on what to do during the lesson, but rather only need to act as a facilitator. This is supported by Yunus and Sukri [11], who stated that self-discovery acts as a very important tool in the classroom because learners will get to learn at their own pace and understand new things based on their own respective abilities. In the current global situation, where the Internet is overtaking human abilities, it is no wonder that young learners are becoming more accepting of the use of e-learning in the teaching and learning of English. To fulfill the needs and wants of the younger generation these days, teachers are now shifting towards other methods of disseminating classroom information. For example, teachers are now more active in using online platforms that focus on game-based learning to help deliver their teaching and learning content in a much more fun and exciting manner. The following section explores in more depth the concept of game-based learning.

*1.2. Game-Based Learning*

Back in the 1980s, the use of games in the teaching and learning of English mainly focused on their effectiveness in the classroom setting. Game-based learning was later found to be effective to help improve learners' learning processes and comprehension of various curricular concepts. In the field of education, specifically the teaching and learning of the English language, game-based learning has been the center of attention as of late, as people are now seeing the high potential of games in improving learners' performance in their education [12]. Over the years, understanding of game-based learning has increased exponentially, alongside the realization that games are able to consolidate the teaching and learning of the English language. When games are used in the classroom, learners can participate more actively [13]. It was also discovered that while playing games, the players would enjoy the opportunity to learn effectively and in a fun manner. Therefore, due to these discoveries over the years, traditional methods of teaching and learning English have become less used in the classroom, as teachers are now moving towards a new era where teaching and learning are preferably done with the implementation of e-learning platforms. An example that is commonly used these days is Quizizz. It is a famous e-learning platform that offers countless quizzes that teachers and learners can use in their daily lessons. The quizzes available on the website can be copied and shared anytime and anywhere, as long as there is an Internet connection. Being a free platform, Quizizz is easily accessible due to its user-friendly interface. Teachers are also able to create their own quizzes based on their own preferences and learners' needs. Therefore, it is clear why teachers would very much prefer to use Quizizz in their teaching and learning processes.

During the recent home-based-learning (HBL) period due to the COVID-19 pandemic, teachers resorted to online platforms to help provide more interesting lessons for their students, instead of just giving them paper-based exercises. One of the most famous platforms used during the HBL period was Quizizz. Due to its ease of use, teachers often send links to quizzes on the platform to their students. At one point in time, it seemed like every teacher was using Quizizz. It is even more attractive to many because it is free, fast, and records of the learners' marks can be tracked and kept. This poses a question for the researchers in this study as to how effective these online platforms are in improving learners' academic performance. This paper aims to investigate what teachers really think about the platform, beyond the ease of sending links due to the effortlessness the platform provides, based on their learners' academic performance and knowledge development. There could be a high possibility that the platform was a trend among teachers, and it may be temporary, or teachers may have seen it as an easy way out instead of having to search for exercises for their students. It is believed that with this systematic literature review of articles related to the use of Quizizz, a conclusion can be drawn to obtain teachers' real perception of the online learning platform Quizizz. In addition to that, the findings from this review will also provide assurance for teachers in the future where there will no longer be any sort of reluctance in using the platform during their lessons postCOVID-19.

## 2. Aim of Current Systematic Review

To address the concern of how teachers view the use of Quizizz in the teaching and learning of the English language, a systematic literature review (SLR) was conducted to understand their perspective and how willing they are to use the platform in the future. As mentioned earlier, this paper will highlight the effectiveness of Quizizz, particularly on learners' academic performance and knowledge development to help other stakeholders realize the potential of the aforementioned online learning platform. This systematic literature review was conducted to answer the following questions:

1. What are teachers' perspectives on the use of Quizizz in the teaching and learning of English?
2. What is teachers' level of willingness to continue using Quizizz in the teaching and learning of English in the future?

## 3. Method

This systematic literature review was conducted using the Preferred Reporting Items for Systematic Review and Meta-Analyses (PRISMA) 2020 checklist. The PRISMA checklist consists of 27 items that help to improve transparency in systematic reviews. This systematic literature review employed the comparative research methodology, specifically descriptive comparison. As this paper aims to describe and explain teachers' perceptions of the use of Quizizz in the teaching and learning of the English language, this paper serves as a systemized endeavor to portray how Quizizz is different and unique when compared to other online learning platforms. By reviewing a selected array of articles related to the use of Quizizz, this paper conceptually analyzes the significant components of how Quizizz contributes to the teaching and learning of the English language, mainly on its effectiveness, feasibility, difficulty, and motivation. Through this method, different researchers' thoughts on these aspects were gathered to juxtapose Quizizz with other online learning platforms. In this review, the study began with the process of identifying articles related to the use of Quizizz in the classroom using the ERIC and Google Scholar databases. The entire process went through four separate phases, mainly the identification phase, screening phase, eligibility phase, and lastly, inclusion phase.

### 3.1. Phase 1: Identification Phase

The databases used for this systematic literature review were ERIC and Google Scholar, and the search range was limited from 2017 to 2021. The Education Resources Information Center (ERIC) serves as a comprehensive digital library that contains various research materials from over 1000 journals worldwide. Meanwhile, Google Scholar is a free web search engine that mostly indexes peer-reviewed scholarly literature across an array of publications. The keywords used when searching for the articles are as seen in Table 1 below. To specify the articles according to the researchers' criteria, additional information was included when searching for the related articles, as shown in Table 2.

**Table 1.** Keywords used to find related articles.

| Databases | Keywords |
|---|---|
| ERIC | Quizizz AND English teaching, Quizizz AND teachers' perspective, Quizizz AND teachers' view, Quizizz AND teachers' motivation, Quizizz AND teachers' willingness, Effectiveness of Quizizz AND teachers |
| Google Scholar | Quizizz AND English teaching, Quizizz AND teachers' perspective, Quizizz AND teachers' view, Quizizz AND teachers' motivation, Quizizz AND teachers' willingness, Effectiveness of Quizizz AND teachers, Impact of Quizizz AND English, Impact of Quizizz AND teachers |

**Table 2.** Inclusion and exclusion criteria.

| Criterion | Inclusion | Exclusion |
|---|---|---|
| Type of article | Journal articles | Book, book chapter, systematic review, proceedings |
| Language | English | Non-English |
| Year | 2017–2021 | <2017 |
| Peer review | Peer-reviewed | Non-peer-reviewed |
| Methodology | Mixed method | Quantitative, qualitative |
| Perspective | Teachers | Students, parents |

### 3.2. Phase 2: Screening Phase

Upon searching for the articles in both ERIC and Google Scholar, duplicates were found and eliminated accordingly. The articles were then examined again to ensure that the remaining ones were in accordance with the criteria set by the researcher.

### 3.3. Phase 3: Eligibility Phase

In the third phase, the gathered articles were checked for their eligibility, where they would have to meet the criteria as stated in the inclusion section of Table 2. It is worth noting that this phase was important to make sure that the data obtained in this study were of good quality and high reliability.

### 3.4. Phase 4: Exclusion Phase

After checking for the articles' eligibility in the third phase, the remaining articles were excluded from this systematic literature review. The excluded articles included those that were not journal articles and were from years before 2017. Non-English-language articles were also excluded. The researchers also made sure that non-peer-reviewed articles were excluded as well, alongside quantitative and qualitative research and articles written on the perspective of students and parents. Just like the eligibility phase, the exclusion phase was also important to make sure the researchers obtained quality data.

Figure 1 shows a clearer view of the entire process from Phase 1 through Phase 4.

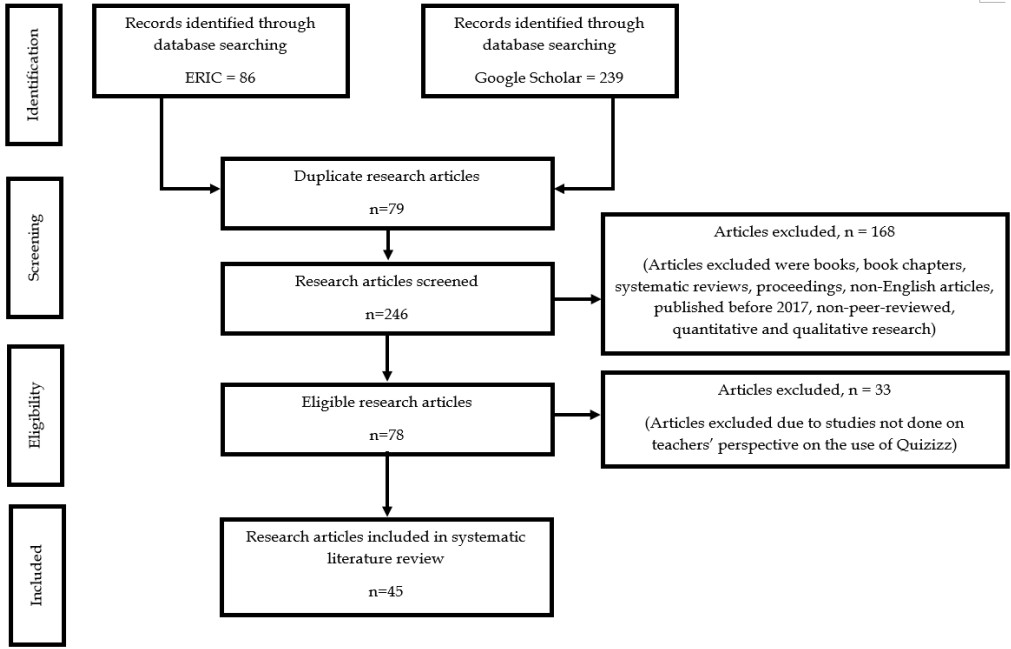

**Figure 1.** Stream chart of the research article selection process.

## 4. Results

In this section, the results from the selection process of the research articles found will be discussed in detail. After the four phases of selecting the eligible articles for review, 45 were found to be suitable to be used in this systematic review due to the nature of the respective studies. From the articles, four main aspects of teachers' perspectives on the use of Quizizz could be seen, mainly, effectiveness [14–42], feasibility [14,15,19,20,22,23,26,27,31,36,38,41,43–48], difficulty [16,17,19,26,28,31,39,41,43,46,48–52], and motivation [14,17,24–26,31,32,35,38,40,41,44,45,47,49,50,53–58]. The results were tabulated as seen in Table 3.

**Table 3.** Aspects of teachers' perspectives on Quizizz.

| Authors | Perspective | | | |
|---|---|---|---|---|
| | **Effectiveness** | **Feasibility** | **Difficulty** | **Motivation** |
| [14] | ✓ | ✓ | | ✓ |
| [15] | ✓ | ✓ | | |
| [53] | | | | ✓ |
| [16] | ✓ | | ✓ | |
| [43] | | ✓ | ✓ | |
| [17] | ✓ | | ✓ | ✓ |
| [44] | | ✓ | | ✓ |
| [54] | | | | ✓ |
| [18] | ✓ | | | |
| [19] | ✓ | ✓ | ✓ | |
| [55] | | | | ✓ |
| [20] | ✓ | ✓ | | |
| [21] | ✓ | | | |
| [49] | | | ✓ | ✓ |
| [22] | ✓ | ✓ | | |
| [23] | ✓ | ✓ | | |
| [45] | | ✓ | | ✓ |
| [50] | | | ✓ | ✓ |
| [24] | ✓ | | | ✓ |
| [46] | | ✓ | ✓ | |
| [25] | ✓ | | | ✓ |
| [26] | ✓ | ✓ | ✓ | ✓ |
| [51] | | | ✓ | |
| [27] | ✓ | ✓ | | |
| [28] | ✓ | | ✓ | |
| [29] | ✓ | | | |
| [30] | ✓ | | | |
| [31] | ✓ | ✓ | ✓ | ✓ |
| [32] | ✓ | | | ✓ |
| [52] | | | ✓ | |
| [56] | | | | ✓ |

**Table 3.** *Cont.*

| Authors | Perspective | | | |
|---|---|---|---|---|
| | **Effectiveness** | **Feasibility** | **Difficulty** | **Motivation** |
| [33] | ✓ | | | |
| [57] | | | | ✓ |
| [47] | | ✓ | | ✓ |
| [34] | ✓ | | | |
| [48] | | ✓ | ✓ | |
| [35] | ✓ | | | ✓ |
| [36] | ✓ | ✓ | | |
| [58] | | | | ✓ |
| [37] | ✓ | | | |
| [38] | ✓ | ✓ | | ✓ |
| [39] | ✓ | | ✓ | |
| [40] | ✓ | | | ✓ |
| [41] | ✓ | ✓ | ✓ | ✓ |
| [42] | ✓ | | | |

Table 4 and Figure 2 below show the breakdown of the number of research articles found on ERIC and Google Scholar in relation to this systematic review.

**Table 4.** Number of articles discussing the different perspectives.

| Perspective | Number of Research Articles |
|---|---|
| Effectiveness | 29 |
| Feasibility | 18 |
| Difficulty | 15 |
| Motivation | 22 |

## Number of journal articles

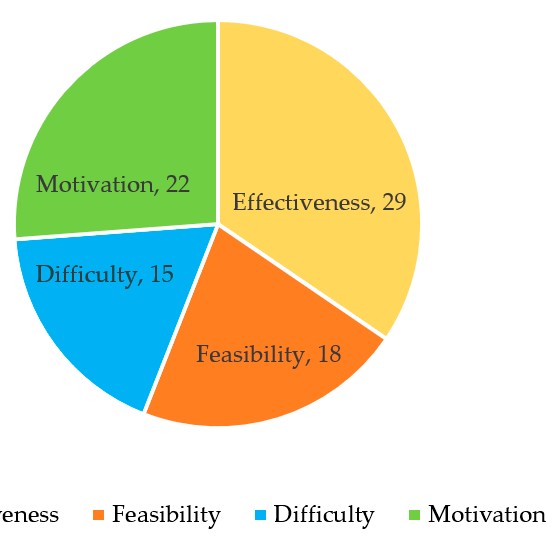

**Figure 2.** The number of journal articles related to the factors studied.

## 5. Discussion

In this section, teachers' perspectives on the use of Quizizz in the teaching and learning of English will be discussed in four main aspects, mainly, effectiveness, feasibility, difficulty, and motivation.

### 5.1. Teachers' Perspective on the Effectiveness of Quizizz

A total of 29 out of the 45 studies discussed teachers' perspectives on the effectiveness of Quizizz in the teaching and learning of English. This represents a vast majority of the research articles found, and after reading through the 29 articles, it is safe to say that teachers mostly agree that the platform is effective in the classroom. Teachers from various countries agreed that Quizizz is effective in many ways, especially in helping learners to learn in a fun, interactive, and meaningful environment [15,19,20,24,25]. Quizizz has a high potential to be exploited in a myriad of ways, including for assessment and classroom exercises [29,31,32,37,38]. Poole and Midura [39] stated in their study that when compared to old traditional methods, Quizizz is better by a large margin due to its ability to attract learners to learn and do better at remembering the important points learned. This is because in the game itself, learners need to earn more points by answering questions correctly. The nature of humans is to win, and this sense of competition among learners will indirectly push them to be better in their studies [26,40–42].

Furthermore, in research conducted by Ahmad et al. [16], it was discovered that teachers find Quizizz very effective because it is highly efficient at promoting students' enthusiasm to learn the language, even more so than other online learning platforms. This is supported by Yunus and Tan [21], who stated that teachers agree that Quizizz is effective because students achieve self-satisfaction while using the platform. In their study, they also learned that Quizizz is effective in the sense that it helps students to learn on their own through memory retention. When the learners have completed one round of the game, they are able to review the questions and do revision on their own. This eventually leads them to remember the lesson content better, and therefore proves that Quizizz is effective in the teaching and learning of the English language [14,17,18,22,34]. An example of the effectiveness of Quizizz can be seen in the research carried out by Grabinski et al. [23], where they used Quizizz as an intervention in the English language classroom. At the end of the research, the researchers discovered that the learners' level of proficiency had increased exponentially with the aid of the online learning platform. Additionally, as English is commonly found to be a difficult subject among learners, Quizizz is able to increase learners' level of intrinsic and extrinsic motivation [33,35,36]. With this, Quizizz eventually helps improve learners' performance in the language due to their eagerness to be actively involved in the learning process [27,28,30].

### 5.2. Teachers' Perspective on the Feasibility of Quizizz

In the research on teachers' perspectives on the use of Quizizz, 18 out of 45 of the articles identified in Phase 4 of this systematic literature review (SLR) discussed teachers' thoughts on the feasibility of Quizizz. This made up 40% of what teachers also considered when deciding whether or not to use Quizizz in their teaching and learning sessions. Quintas-Hijos et al. [44] stated that teachers consider Quizizz quite feasible because using it is convenient for all users. However, they still need to be connected to the Internet in order to participate in the sessions. This was a concern for many teachers, because Internet connectivity is still a problem in many areas, including those in suburban areas [36,43,47,48]. Most teachers agree that Quizizz is indeed effective for the teaching and learning of the English language, but that it is still important to make sure that the learners have equal opportunities to learn, including having equal access to the Internet [22].

In using Quizizz, learners would not have much problem in navigating around the website. Its interface, which is user-friendly, helps learners to access different functions without much hassle. With this, learners who have limited Internet access would not waste much time or Internet quota on navigating around aimlessly [19,20,44]. In the teaching and

learning process, students only need a game code from their teachers in order to access a game that can be played in real-time mode or student-paced mode [26,41]. This provides a fairer chance for learners to be active in the learning process because they have the freedom to choose when and where they would like to complete the quizzes given by the teacher, thus making Quizizz more feasible for learners. Overall, teachers agree that Quizizz is easy and convenient to use for everyone, including learners and teachers themselves.

*5.3. Teachers' Perspective on the Difficulty of Using Quizizz*

Quizizz is an online platform that is free for everyone—not only teachers, but also learners. With only a click of a button, users can sign up for free and have unlimited access to various quizzes that suit their respective levels. The quizzes available on the platform can be used by teachers at their discretion and teachers can also create or design their own quizzes any time they wish. From the 45 research articles identified in this SLR, 15 discussed teachers' perspectives on the difficulty level of using Quizizz. Teachers understood that with enough practice, they will be accustomed to the various features of the website [28,51]. However, it is still a concern to many who believe students might face difficulties using the platform. Degirmenci [31] stated that there will be times when learners will face difficulties using Quizizz due to various factors such as Internet connection and device availability. This is supported by Poole and Midura [39], Moakofhi et al. [48], and Weng et al. [52], who claimed that Internet connection hinders the maximum exploitation of various online platforms such as Quizizz. As mentioned earlier, this causes inequality in education, especially among poorer students. This has become a concern for teachers, many of whom do not incorporate online platforms into their lessons because they realize that not all learners have easy access to the Internet. Furthermore, even those who do have access to the Internet may not necessarily have it at a low price. Many would have to save on their Internet quota and thus, they would rather not take part in online sessions such as Quizizz [19,46,49,50].

Another aspect of difficulty found in these 15 research articles was the difficulties faced by teachers in creating or using the quizzes on Quizizz. Senior teachers mentioned that they are not interested in using Quizizz because they do not know how to use the platform entirely. In addition, the factors of age and time have become a common reason given by teachers to not use Quizizz in teaching and learning sessions in the classroom [16,17,43]. However, these difficulties can be overcome easily with some practice or short courses that can be done even at the school level. It is important to understand that with exposure, learners will be used to such platforms and thus, they will be more ready for the future [41].

*5.4. Teachers' Perspectives on How Quizizz Motivates Learners*

Teachers' perspectives on the use of Quizizz in the teaching and learning of English could also be seen through how it motivates learners to learn the language. In the learning process, motivation plays a huge role. Without motivation, learners would not see the reason why they should learn the subject, in this case, the English language. As asserted by Kayseroglu and Samur [24], motivation creates a positive perception and through this, learners will be more willing to be part of the teaching and learning process. Out of the 45 research articles, 22 of them discussed teachers' perspectives on how Quizizz motivates learners.

In a research conducted by Bereczki and Karpati [49], the teachers who were interviewed during the data collection process mentioned that they see huge potential in Quizizz being used in the classroom for a long time because it motivates learners to learn more effectively. Learners would want to win the games they participate in and therefore, through constant playing of the games, they will remember the lesson content even without realizing it. Learners are also motivated to learn when they use Quizizz because they are learning in a fun, comfortable, and conducive environment with a little bit of friendly competition [35,38,47,57,58].

Furthermore, in a study done by Seixas et al. [25], it was found that 87% of the teachers who were the research participants agreed that their learners showed positive progress in their level of proficiency after using Quizizz in the classroom. The current generation is a generation that is very used to a gamified environment. This clearly shows that if teachers continue to use traditional methods in their teaching, it will be redundant to the learners because they are not used to such methods. As agents of change, teachers must constantly be up to date with the latest teaching and learning tools so that they can make sure that the learners are effectively learning no matter where they are, be it in urban or rural areas. Thus, it is safe to say that generally, teachers agree that Quizizz is a good tool to be used in the teaching and learning of the English language, as it is able to motivate learners to learn and improve positively [31,32,44,45,54,55].

## 5.5. Teachers' Willingness to Continue Using Quizizz in the Classroom

Meanwhile, the results on teachers' willingness to use Quizizz in their classes could also be seen in the selected articles. Most teachers in the research conducted had a positive outlook on Quizizz and were willing to incorporate the platform in the future [14–17,19,21–25,27,28,30–38,40,41,43–46,48,50,51,53–55,57,58]. The findings on teachers' willingness to use Quizizz, which is related to this review's second research question, are shown in Table 5.

**Table 5.** Findings on teachers' willingness to use Quizizz.

| Authors | Willingness to Use Quizizz | | |
|:---:|:---:|:---:|:---:|
| | Willing to Use | Not Willing to Use | Not Discussed |
| [14] | ✓ | | |
| [15] | ✓ | | |
| [53] | ✓ | | |
| [16] | ✓ | | |
| [43] | ✓ | | |
| [17] | ✓ | | |
| [44] | ✓ | | |
| [54] | ✓ | | |
| [18] | | | ✓ |
| [19] | ✓ | | |
| [55] | ✓ | | |
| [20] | | | ✓ |
| [21] | ✓ | | |
| [49] | | ✓ | |
| [22] | ✓ | | |
| [23] | ✓ | | |
| [45] | ✓ | | |
| [50] | ✓ | | |
| [24] | ✓ | | |
| [46] | ✓ | | |
| [25] | ✓ | | |
| [26] | | | ✓ |
| [51] | ✓ | | |

**Table 5.** *Cont.*

| Authors | Willingness to Use Quizizz | | |
| --- | --- | --- | --- |
| | **Willing to Use** | **Not Willing to Use** | **Not Discussed** |
| [27] | ✓ | | |
| [28] | ✓ | | |
| [29] | | | ✓ |
| [30] | ✓ | | |
| [31] | ✓ | | |
| [32] | ✓ | | |
| [52] | | ✓ | |
| [56] | | | ✓ |
| [33] | ✓ | | |
| [57] | ✓ | | |
| [47] | | | ✓ |
| [34] | ✓ | | |
| [48] | ✓ | | |
| [35] | ✓ | | |
| [36] | ✓ | | |
| [58] | ✓ | | |
| [37] | ✓ | | |
| [38] | ✓ | | |
| [39] | | | ✓ |
| [40] | ✓ | | |
| [41] | ✓ | | |
| [42] | | ✓ | |

As seen in Table 4, 35 out of 45 of the identified research articles mentioned that teachers are willing to continue using Quizizz in their future lessons. Teachers see Quizizz as a very useful tool as it creates a fun learning environment for the learners. It is no wonder that learners become bored in the classroom when their teachers only use old methods to teach, especially when the process is only one-way, from the teacher to the learners [32,37]. Learners should be given the autonomy to take control of their own learning. Being an independent learner permits them to be more responsible, aside from motivating learners to be better, and this will lead them to achieve better results and proficiency in the language [59,60]. With this in mind, it is no question that teachers would definitely continue to use Quizizz in the teaching and learning of English in school.

## 6. Conclusions

This systematic literature review analyzed teachers' perspectives on the use of Quizizz in the teaching and learning of the English language. Their perspectives were analyzed from four different aspects, mainly, effectiveness, feasibility, difficulty, and motivation. The results show that teachers look highly upon Quizizz and its implementation in the classroom because it brings many benefits to the learners, including improvement in language proficiency and learning abilities. As a final result, based on the review of all the articles, teachers definitely view Quizizz as a platform that is effective, feasible, easy to use, and motivating for their learners, thus making it an online learning platform that is able to facilitate learners' academic achievement and knowledge development. The nature of Quizizz helps to create a fun learning environment and undeniably, this will

affect learners' performance in school, where they will be more motivated to learn and be better than their peers. As asserted by Papadakis [61], the demand for professionals with great computing skills is growing and it is pivotal to develop computational thinking skills to produce younger generations that can succeed in our complex and technological culture in the future. Aside from that, this systematic literature review also analyzed teachers' willingness to continue using Quizizz in their classrooms in the future. Out of the 45 research articles gathered, 35 mentioned that the teachers in their respective studies were willing to continue using Quizizz in the future. This evidently shows that Quizizz has left quite an impression on teachers from various backgrounds, and they are now more accepting of the platform. With teachers believing in the effectiveness of Quizizz, as agents of change, they will be able to motivate various members of the community to be more accepting of the use of Quizizz instead of just viewing it as a gaming platform with no benefits. It is high time for all stakeholders of the education system to realize and understand the potential of digital games in enhancing education for learners of various levels, especially in the current digital environment we are now living in [62]. Therefore, the research questions of this current study are answered, and it is safe to say that from the perspective of teachers, Quizizz is in fact effective, feasible, easy to use, and motivating for all learners, thus proving that Quizizz has high potential to improve academic performance and facilitate knowledge development.

## 7. Challenges and Limitations

In this study, there were limitations that would be worth addressing in the future in case there are researchers who would like to investigate this topic further. This study only reviewed 45 articles from ERIC and Google Scholar. However, there might be other articles from other databases like SCOPUS and WoS. Due to time constraints, the authors of this current study only focused on articles from ERIC and Google Scholar. Expanding the research to more views from various scholars would be interesting, as the results may further strengthen the findings of this study, or it could even show different results with different schools of thought. The second and last limitation of this research is that the study did not get to meet the authors of the various articles reviewed in this systematic review. A suggestion would be to meet the authors, whether virtually or in person, so that a more in-depth investigation could be conducted. However, as mentioned earlier, time constraints limited the authors of this study to not be able to speak directly with the researchers.

## 8. Implications and Recommendations

This systematic literature review has brought a new view to the use of Quizizz in the field of education, particularly in the teaching and learning of the English language. Teachers have a positive outlook on the use of Quizizz in the classroom. However, through this research and after reading the identified articles, it has come to our attention that teachers have a big concern for Internet connection, which is directly linked to learning opportunities among learners. As the world heads towards major globalization, Internet connection is still hard to obtain in many parts of the world. As many are quickly moving towards a modern world where technology is now a huge part of our lives, a big number of citizens are being left behind. Those left behind are often forgotten and their lives are often taken for granted, including basic necessities like education. Stakeholders in the field of education should play more active roles in making sure all students, regardless of race, socio-economic status, or origin, have equal access to education and that no one gets left out. If it is our dream to have a country with proper education, equality must be available and enjoyed by all. As a proposal for future studies, the issue of the implementation of game-based learning should be looked into in more detail, particularly in terms of Internet connection and device availability. More importantly, the way forward to help overcome these problems should be discussed as well. Future researchers should conduct more research on the use of online learning platforms to provide more awareness of their effectiveness in promoting learning among students of all ages. Only with sufficient studies

being conducted can the community see the benefits that platforms like Quizizz have to offer for the education system globally.

**Author Contributions:** All authors contributed to several aspects of the study, specifically, conceptualization, T.M.L. and M.M.Y.; methodology, M.M.Y.; validation, M.M.Y.; formal analysis, T.M.L.; investigation, T.M.L.; resources, M.M.Y.; data curation, T.M.L. and M.M.Y.; writing—original draft preparation, T.M.L.; writing—review and editing, T.M.L. and M.M.Y.; supervision, M.M.Y.; project administration, T.M.L.; funding acquisition, M.M.Y. All authors have read and agreed to the published version of the manuscript.

**Funding:** This research was funded by Universiti Kebangsaan Malaysia under research grant number GG-2020-024 and the APC was funded by Universiti Kebangsaan Malaysia.

**Institutional Review Board Statement:** Not applicable.

**Informed Consent Statement:** Not applicable.

**Data Availability Statement:** Not applicable.

**Conflicts of Interest:** The authors declare no conflict of interest.

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
