# Peer review of "Teachers’ Perception towards the Use of Quizizz in the Teaching and Learning of English: A Systematic Review"

_sustainability, doi:10.3390/su13116436_

Round 1

Reviewer 1 Report

This is informative, interesting and well-argued analytical article.

The only revision would be adding methodology employed early in the manuscript. Specifically, what methodology is being employed?  Is it is comparative research methodology, or discourse analysis? 

Reviewer 2 Report

My personal opinion is that after reading the paper, the manuscript is of potential interest to the readership of this journal, but there are significant issues that must be addressed before the article could be published:

In general:

  1. Background – Expand a little more to highlight the research problem to highlight the study's need.
  2. Methodology - expand a little more. Add analysis methods.
  3. Contribution: It would be a good paper if it did look at the research impact on the community.
  4. Findings: Should align with the study goal. 
  5. Recommendations: Expand a little more.
  6. Recommendation for Researchers

Introduction

A concise introduction to enable the reader's understanding of the research problem.

  • Introduce the paper describing what the paper is about.  Expand to emphasize the problem leading to a clear set of research questions and objectives the research addresses.
  • Give readers a one-line preview of the other sections of the paper.

Literature review

Used sub-headings to organize topics. Some critical studies are not included. The paper should relate coherently and convincingly with issues of real-world significance. This is a crucial phase contributing to research design. The theoretical framework emerging from the literature review could research questions and points of emphasis.

Suggestions

  • Include a few introductory lines to indicate what the review will cover, outlining the purpose and scope.
  • Consider summarising the text based on the study purpose.
  • Focus more on the empirical studies' backgrounds.
  • Add more information to enable readers' understanding of the authors' view.
  • You are encouraged to write concisely. The text can be reduced significantly.

Findings and discussion

Needs clear and comprehensive explanations to assist readers' understanding.

Challenges

How were these reported in the discussion?

Conclusion

The conclusion falls short of providing sufficient information that would allow a reader to understand the contribution of this research.  What was found? I would expect the conclusion to refer back to the research questions.

Limitations – There is no mention of the limitations of this study.

Summary

 The study presented an important topic that would be of interest to the readership of this journal. Most research is needed, including the international audience. However, the research does not match what is wanted in its current form, and the surveys do not provide relevant data. It is missing a level of detail needed to understand the study result, the impact of the results, and the research contribution. Perhaps the authors are likely close to the topic they are skipping over details that they know, but the reader would not.

Overall, the paper requires more information and focus. The areas requiring attention are highlighted in the individual sections. 

In summary, the paper needs

  • a re-write of the abstract to give a good summary of the paper and mention the key concepts.
  • Expanding the introduction by clearly stating the research problem to suitably inform the reader.
  • A synthesised and structured critique of the literature.
  • clarifying the research procedures with an adequate explanation of the methods.
  • Improving the survey instrument – construction of the questionnaire and the validity tests.
  • Expanding the discussion to allow writing a well-developed conclusion summarising the entire paper. The outcomes should be discussed in relation to the existing research.
  • emphasizing the significance of the research - a clear showing of how the findings contribute to new knowledge.
  • Using results to support the claims in conclusion adequately, and how the results of the research can be used for future research
  • A more recent bibliography is necessary. Remove all outdated references! Furthermore, the reference list of new publications is a little bit weak. There are not enough studies from Europe or western countries. Before I can make a final decision on the paper, please refer to more references. It is suggested that the author(s) can consider the following papers related to the use of gamification etc. to strengthen the background and conclusions of the study:
  • Papadakis, S. (2020). Evaluating a game-development approach to teach introductory programming concepts in secondary education. International Journal of Technology Enhanced Learning, 12(2), 127-145.
  • Papadakis, S.; Trampas, A.; Barianos, A.; Kalogiannakis, M. and Vidakis, N. (2020). Evaluating the Learning Process: The “ThimelEdu” Educational Game Case Study. In Proceedings of the 12th International Conference on Computer Supported Education - Volume 2: CSEDU, ISBN 978-989-758-417-6, pages 290-298. DOI: 10.5220/0009379902900298

In general, the English in the present manuscript is of publication quality and requires minor improvement. Please carefully proof-read spell check to eliminate grammatical errors

Plagiarism check results:

/* Similarity check with iThenticate revealed a similarity index of 7%, which is considered appropriate. A maximum of around 60 quoted words is accepted per paper. There are no papers with over 60 words. No previously copyrighted material was used.  

In preparing a revised manuscript, please also include a table of how you have responded to each of the issues listed above point by point.

Dear AUTHOR, summarizing my feedback, I expect your contribution to be highly valued by the journal's readers if you improve it according to the review statements. Please resubmit for review within a month at the latest.

I look forward to receiving your revised manuscript shortly.

With best regards,

Reviewer 3 Report

As the authors indicate, the four skills on which English language teaching and learning focuses are listening, speaking, reading and writing. It would have been very interesting if the meta-analysis had included teachers' perceptions of the use of Quizizz to achieve these skills, and thus English language proficiency, and not just focused on perceptions of effectiveness, feasibility, difficulty and motivation.

Without a clear prior definition of effectiveness, feasibility or viability, difficulty and motivation, it may happen that, in the documents consulted, the authors used one of these terms thinking indistinctly of another, I mean, it may happen that the authors reviewed did not discriminate between the terms: effectiveness, feasibility or viability, difficulty and motivation. In this sense, in the discussion itself, in the document presented, feasibility sometimes seems to be mixed up with difficulty and, in some cases, effectiveness and motivation, which are undoubtedly related variables or concepts.

The document moves from the introduction and objectives directly to the methodology, without including a minimum theoretical framework with several sections. One of the sections could be, for example, the teaching and learning possibilities of Quizizz for acquiring the necessary skills in English or why are other, probably more widespread, platforms, for example Kahoot, not used?

The second research question does not make sense without interviewing the reviewed authors to obtain more precise qualitative results. The results in table 5 are obvious before doing the study, in this sense, those who publish about their experiences with a support tool, in this case Quizizz, is because their results have been satisfactory in one way or another, on the other hand, many authors do not feel the need to state in their studies whether or not they will continue to use a particular teaching-learning tool.

Round 2

Reviewer 3 Report

The article has improved with your new contributions